# $^1$H, $^{13}$C and $^{15}$N resonance assignment of the SARS-CoV-2 full-length nsp1 protein and its mutants reveals its unique secondary structure features in solution

**Tatiana Agback[1], Francisco Dominguez[2], Ilya Frolov[2], Elena I. Frolova[2], Peter Agback[1] ***

**1** Department of Molecular Sciences, Swedish University of Agricultural Sciences, Uppsala, Sweden,
**2** Department of Microbiology, University of Alabama at Birmingham, Birmingham, AL, United States of America

* peter.agback@slu.se

**Data Availability Statement:** All 1H, 15N and 13C chemical shifts of the full-length of SARS-CoV-2 nsp1 protein at pH 7.5 and at two temperatures, 298K and 308K, have been deposited in

## Abstract

Structural characterization of the SARS-CoV-2 full length nsp1 protein will be an essential tool for developing new target-directed antiviral drugs against SARS-CoV-2 and for further understanding of intra- and intermolecular interactions of this protein. As a first step in the NMR studies of the protein, we report the $^1$H, $^{13}$C and $^{15}$N resonance backbone assignment as well as the Cβ of the apo form of the full-lengthSARS-CoV-2 nsp1 including the folded domain together with the flaking N- and C- terminal intrinsically disordered fragments. The 19.8 kD protein was characterized by high-resolution NMR. Validation of assignment have been done by using two different mutants, H81P and K129E/D48E as well as by amino acid specific experiments. According to the obtained assignment, the secondary structure of the folded domain in solution was almost identical to its previously published X-ray structure as well as another published secondary structure obtained by NMR, but some discrepancies have been detected. In the solution SARS-CoV-2 nsp1 exhibited disordered, flexible N- and C-termini with different dynamic characteristics. The short peptide in the beginning of the disordered C-terminal domain adopted two different conformations distinguishable on the NMR time scale. We propose that the disordered and folded nsp1 domains are not fully independent units but are rather involved in intramolecular interactions. Studies of the structure and dynamics of the SARS-CoV-2 mutant in solution are on-going and will provide important insights into the molecular mechanisms underlying these interactions.

## Introduction

Within the recent 1.5 years, the severe acute respiratory syndrome coronavirus 2 (SARS-CoV-2) has spread world-wide and devastated the economies of essentially all countries [1, 2]. To date, more than one hundred million people have contracted the disease that led to more than 3 M deaths (https://www.worldometers.info/coronavirus). Despite the enormous public health threat of this and previous CoV infections, no efficient therapeutic means have been developed

BioMagResBank, (http://www.bmrb.wisc.edu) under the accession 50915. Released upon publication of manuscript.

**Funding:** This work was supported by Swedish Foundation for Strategic Research grant ITM17-0218 to P.A. (https://strategiska.se/), Public Health Service grant R21AI146969 to I.F. (https://www.niaid.nih.gov/) and UAB Research Acceleration Funds to E.F. and I.F. (https://www.uab.edu/home/) The funders had no role in study design, data collection and analysis, decision to publish, or preparation of the manuscript.

**Competing interests:** The authors have declared that no competing interests exist.

against coronaviruses (CoV) before the COVID-19 pandemics. One of the major reasons for this was a lack of detailed knowledge of the mechanism of CoV replication and their interactions with host cells.

SARS-CoV-2 is a member of the *Betacoronavirus* (β-CoV) genus along with other highly pathogenic respiratory viruses, such as SARS-CoV-1 and MERS-CoV (Middle Eastern respiratory syndrome virus). These viruses have similar genome and replication strategies but differ in their pathogenicity for humans. Similar to other β-CoVs, the SARS-CoV-2 genome (G RNA) is represented by a single-stranded RNA of positive polarity of ~30 kb in length [3–5]. It mimics the structure of cellular mRNAs in that it has a Cap and a poly(A)-tail at the 5' and 3' termini, respectively. Upon delivery into the cells, the G RNA is directly translated into two very long polyproteins, which are encoded by the overlapping ORF1a and ORF1b. The latter polyproteins are enzymatically processed into individual nonstructural proteins nsp1-to-16 by the encoded protease activities. These nsps represent viral components of the replication complexes and are also involved in modification of the intracellular environment to promote efficient viral replication. As in the case of other β-CoVs, the SARS-CoV-2-specific nsp1 protein plays indispensable roles in these processes [6–8]. First of all, it is a key player in downregulation of cellular translation and is a major β-CoV-specific virulence factor [6, 9, 10]. It interacts with the 40S ribosomal subunit, blocks the RNA channel and inhibits initiation of translation of cellular, but not viral, RNA templates [6, 8, 11–17]. SARS-CoV-1 and MERS nsp1 proteins are also indirectly involved in endonuclease degradation of cellular mRNAs; however, the mechanism of this function remains to be determined [18, 19]. It is still unknown whether SARS-CoV-2 nsp1 can mediate degradation of cellular RNAs. Nsp1 of both SARS-CoV-1 and SARS-CoV-2 were also implicated in inhibition of nuclear-cytoplasmic traffic, albeit by different mechanisms [20, 21]. The above activities appear to play critical roles in the downregulation of the innate immune response that can mount during SARS-CoV-2 infection, and thus, control the infection spread. Nsp1 proteins of β-CoVs also facilitate cell cycle arrest, which is clearly detectable during viral infection and nsp1 expression [9, 22]. Importantly, the previous studies demonstrated that the deletion of nsp1 gene in the genome of other β-CoVs makes them nonviable [23]. This strongly indicated the direct involvement of the latter protein in genomic RNA replication and/or synthesis of the subgenomic RNAs, which encode viral structural and accessory proteins. Interestingly, point mutations or small deletions in nsp1 can independently prevent either inhibition of cellular translation or viral replication [22–27].

Thus, the accumulated data suggest that nsp1 plays important roles in CoV replication and pathogenesis. It exhibits multiple activities and likely interacts with a variety of viral and cellular proteins and organelles. Understanding of the molecular mechanisms of these interactions is critical for development of live attenuated vaccines and therapeutic means against SARS-CoV-2 infection. Further dissection of multiple nsp1 functions in viral replication and pathogenesis requires the detailed knowledge of the dynamic structure of nsp1. To date, the data about the structure of β-CoV nsp1 remain very fragmented. This is a relatively small 19.8 kDa protein. It contains an N-terminal structured domain (amino acids 13 to 125 in SARS-CoV-2 nsp1), which was proposed to be critical for degradation of cellular mRNAs. The first 12 residues and the C-terminal fragment (residues 126–180) in SARS-Cov-2 nsp1 are predicted to be intrinsically disordered. However, the last 26 amino acid long peptide in this C-terminal fragment has been shown to fold into two short α-helixes upon binding to the 40S ribosome subunit [7, 8, 17]. The structure of the folded N-terminal domain of SARS-CoV nsp1 has been determined by NMR (PDB:2HSX), and two X-ray structures of the folded domain of SARS-CoV-2 nsp1 have been recently published (PDB:7K7P and 7K3N) [28–30]. The N-terminal nsp1 domains of both viruses have similar folds. The important difference was found to be the presence of an additional small β-strand (residues 95–97) in SARS-CoV-2-specific nsp1. No

structure of the full-length proteins containing the disordered C-terminal domain is available for the nsp1 of any β-CoV.

The flexibility of the disordered C- and N-terminal fragments in the SARS-CoV-2 nsp1 and their interactions with the folded domain may play the critical role(s) in protein functions. Solution NMR is the method of choice for studying such flexible regions in proteins. Most of the currently available NMR approaches and protocols are focused on elucidating the structure of either fully folded proteins (FP), which complement the crystallographic data, or the intrinsically disordered protein (IDPs). Complete backbone and side chain resonance assignment of NMR spectra of large proteins, containing both folded and disordered domains, is still a challenge due to (a) a high degree of divergence in the conformational flexibility characteristics of disordered and folded domains and (b) the reduced frequency dispersion observed in the NMR spectra in the $^1$H dimension for disordered regions. Recently, a near-complete backbone resonance assignment of the SARS-CoV-2 nsp1 was reported [31]. The latter protein was analysed in an acidic buffer, pH 6.5.

As a first step towards characterizing the structure and dynamics of the full-length SARS-CoV-2 nsp1 in neutral buffer by NMR spectroscopy, we herein report the almost complete $^1$H, $^{13}$C and $^{15}$N backbone and $^{13}$Cβ side chain assignment of the wild type protein and two of its mutants: a single mutant H81P and a double mutant K129E, D48E. This assignment has been evaluated by additionally observing selectively chosen type of amino acids (MUSIC-type of experiments). To overcome ambiguities in assignment in the crowded areas of the full-length SARS-CoV-2 nsp1 spectra, we additionally compared the assigned resonances with those corresponding to the single and double mutants of nsp1. Based on this data, the secondary structure of the full-length SARS-CoV-2 nsp1 in solution was derived and, additionally, the protein flexibility was evaluated. These data provide a structural basis for further understanding of intra- and intermolecular interactions of the SARS-CoV-2 nsp1.

## Material and methods

### Preparation of $^{15}$N, $^{13}$C labeled SARS-CoV-2 nsp1 proteins

Nucleotide sequence of the full-length wild type nsp1 was amplified by PCR from recombinant cDNA of SARS-CoV-2 Wuhan-Hu-1 strain (NC_045512.2) using primers:

NSP1-Dir `CCACTGGTCTCAAGGTGGTATGGAGAGCCTTGTCCCTGG`

NSP1-Rev `CCACACTCGAGTTATTACCCTCCGTTAAGCTCACGC`.

The PCR product was cloned into pE-SUMOpro-3 plasmid (LifeSensors Inc) between Bsa I and Xho I restriction sites. The synthetic gene blocks encoding mutant nsp1 were ordered from Integrated DNA Technologies and cloned into pE-SUMOpro-3 plasmid (LifeSensors Inc) between Eco RI and Xho I restriction sites. Plasmids encoding SUMO-nsp1 proteins were transformed into *E. coli* strain Rosetta2(DE3)pLacI (Novagen), and proteins were produced in the M9 media supplemented with 2 g/L [$^{15}$N]NH$_4$Cl (Cambridge Isotope Laboratories) and 3 g/L d-[$^{13}$C$_6$]glucose (Cambridge Isotope Laboratories). The expression was induced by 1 mM IPTG after cells reached the density of ~2 OD$_{600}$. Then cells continued to grow at 37˚C for 3–5 h. Freshly prepared or frozen cell pellets were lysed in Emulsiflex B15 (Avestin). The lysates were loaded on HisTrap HP column (GE Healthcare) and after extensive washing the recombinant proteins were eluted by imidazole gradient. Fractions containing SUMO-HVD nsp1 proteins were combined, and His-SUMO tag was cleaved with Upl1 protease. After cleavage, the proteins were diluted to contain 25 mM NaCl and further purified on Resource Q column (GE Healthcare). Size exclusion chromatography on a HiLoad Superdex 75 16/600 column (GE Healthcare) in NMR buffer was used as a final purification step. Fractions containing pure proteins were combined and concentrated.

The protein purities and identities were confirmed by SDS-PAGE and mass spectrometry (for unlabeled protein), respectively. Final proteins contained an extra glycine at the N-terminus, which was required for SUMO cleavage. Protein concentrations were determined on 280 nm using extinction coefficients, which were determined by ProteinCalculator v3.4 (http://protcalc.sourceforge.net/).

## Preparation of NMR samples

All NMR experiments were performed in a buffer containing 20 mM HEPES pH 7.5, 100 mM KCl (with or without 2.5 mM $MgCl_2$), 1 mM $NaN_3$, 10 (v/v) % $D_2O$ and 0.1 mM DSS (4,4-dimethyl-4-silapentane-1-sulfonic acid) as an internal $^1H$ chemical shift standard. The protein concentration were about 0.4 mM and spectra were acquired in a 5 mm tube (final volume of 0.45mL). $^{13}C$ and $^{15}N$ chemical shifts were referenced indirectly to the $^1H$ standard using a conversion factor derived from the ratio of NMR frequencies [32].

To perform 3D $^{13}C$-HSQC type experiments, the nsp1 sample was lyophilized and dissolved in $D_2O$ in a volume of 0.45 mL. To confirm that the protein's structure was not affected by lyophilization, sample was lyophilized again, dissolved in $H_2O$ and 1D $^1H$ spectrum was compared to the original spectrum.

## NMR experiments

NMR experiments were acquired on a Bruker Avance III spectrometers operating at 14.1 T, corresponding to 600 MHz, equipped with a cryo-enhanced QCI-P probe. The assignment of the backbone and side chains resonances was based on a set of 3D TROSY or HSQC triple resonance experiments from the Bruker library. The summary of the performed experiments and the key parameters are presented in Table 1. To increase resolution in the indirect dimensions and reduce acquisition time of the 3D experiments, the iterative non-uniform sampling protocol (NUS) [33] was used in the experiments comprising: TROSY-HNCO, TROSY-HNCA and TROSY-HN(CO)CA, TROSY-HN(CA)CO, TROSY-HN(CO)CACB, TROSY HNCACB, H(CC)(CO)NH, TOCSY- $^{15}N$ HSQC, HCACO and TOCSY-$^{13}C$ HSQC experiments. NUS points sampling schedules applied to the 3D experiments are listed in Table 1.

To assign Hα proton resonances and for assessment of NOE contacts, additional NOESY $^{15}N$-HSQC, NOESY-$^{13}C$- HSQC [34–36] spectra were collected. The quality of the spectra allowed for assignment of the Hα protons and made HACACO experiment superfluous.

To verify assignment of the amino acids located in the disordered fragments of SARS-CoV-2 nsp1, $^{13}C$ observed CON experiment with IPAP scheme for virtual decoupling [37, 38] was used to correlate $^{15}N$ with 13C' resonances.

We also performed TROSY type MUSIC experiments with semi-constant time acquisition period in the indirect dimension to identify different type of amino acids such as Ser, Glu, Ala, Asp, Gln, Asn and their respective n+1 residues. Key parameters of the experiments used in this study are presented in Table 1. All TROSY-type MUSIC pulse sequences and the setting details have been fully described elsewhere [39].

Due to the differences in the relaxation characteristics of the folded domain and the disordered fragments of SARS-CoV-2 nsp1, several experiments were performed at two temperatures: 298 K and 308 K.

Data were processed by Topspin 4.0.6 (Bruker) using linear prediction and assigned using CcpNmr Analysis 2.4.2 [40].

The chemical shifts of the full-length SARS-CoV-2 nsp1 were analyzed with TALOS-N software [41]. As input for TALOS-N analysis, the experimentally derived chemical shifts of $^1HN$, $^{15}N$, $^{13}Cα$, $^{13}Cβ$, $^{13}C'$ and $^1H^α$ nuclei for every amino acid were used. In case of absence of

**Table 1. List of NMR experiments and the main parameters used to perform the sequence-specific assignment of the backbone and side chain resonances of the full-length SARS-CoV-2 nsp1 protein.**

| Experiments | Maximum evolution time, ms. | | | scans | NUS, points | NUS, % | time |
|---|---|---|---|---|---|---|---|
| | F3 | F2 | F1 | | | | |
| $^1$H-$^{15}$N HSQC[a] | 106.5($^1$H) | 164.4($^{15}$N) | - | 8 | - | - | 2h10m |
| TROSY-HNCO[a] | 106.5($^1$H) | 12.3($^{15}$N) | 33.1($^{13}$C) | 4 | 900 | 30 | 4h51m |
| TROSY-HN(CA)CO[a] | 106.5($^1$H) | 12.3($^{15}$N) | 21.2($^{13}$C) | 32 | 576 | 30 | 1d1h |
| TROSY-HNCA[a] | 106.5($^1$H) | 12.3($^{15}$N) | 16.5($^{13}$C) | 8 | 900 | 30 | 9h40m |
| TROSY-HN(CO)CA[a] | 106.5($^1$H) | 12.3($^{15}$N) | 16.5($^{13}$C) | 8 | 900 | 30 | 19h38m |
| TROSY-HNCACB[a] | 106.5($^1$H) | 10.2($^{15}$N) | 8.3($^{13}$C) | 32 | 750 | 30 | 1d8h |
| TROSY-HN(CO)CACB[a] | 106.5($^1$H) | 10.2($^{15}$N) | 12.4($^{13}$C) | 32 | 1125 | 30 | 2d1h |
| 3D H(CC)(CO)NH | 106.5($^1$H) | 10.7($^{15}$N) | 3.3($^1$H) | 16 | 208 | 25 | 4h39m |
| 3D $^1$H–$^{15}$N NOESY[a] | 106.5($^1$H) | 12.3($^{15}$N) | 6.6($^1$H) | 16 | - | - | 1d20h |
| 3D DIPSI2- $^{15}$N HSQC | 106.5($^1$H) | 13.1($^{15}$N) | 9.8($^1$H) | 16 | 902 | 30 | 20h8m |
| 3D $^1$H–$^{13}$C NOESY[b] | 155.1($^1$H) | 3.3($^{13}$C) | 15.1($^1$H) | 8 | - | - | 1d22h |
| 3D HCACO[b] | 426.5($^1$H) | 9.9($^{13}$C) | 19.8($^{13}$C) | 8 | 900 | 25 | 1d |
| 3D MLEV17-$^{13}$C HSQC[b] | 155.1($^1$H) | 7.4($^{13}$C) | 15.1($^1$H) | 8 | 2250 | 25 | 1d1h |
| 2D CON IPAP | - | 169.6($^{13}$C) | 44.7($^{15}$N) | 32 | - | - | 2h53m |
| TROSY-MUSIC[c] (S+1) | 106.5($^1$H) | 16.4($^{15}$N) | - | 64 | - | - | 1h47m |
| TROSY-MUSIC[c](S+1, S) | 106.5($^1$H) | 16.4($^{15}$N) | - | 64 | - | - | 1h45m |
| TROSY-MUSIC[c](E+1) | 106.5($^1$H) | 16.4($^{15}$N) | - | 128 | - | - | 3h45m |
| TROSY-MUSIC[c](E+1, E) | 106.5($^1$H) | 16.4($^{15}$N) | - | 128 | - | - | 3h40m |
| TROSY-MUSIC[c](A+1) | 106.5($^1$H) | 16.4($^{15}$N) | - | 128 | - | - | 3h34m |
| TROSY-MUSIC[c](A+1, A) | 106.5($^1$H) | 16.4($^{15}$N) | - | 128 | - | - | 3h30m |
| TROSY-MUSIC[c](D+1) | 106.5($^1$H) | 16.4($^{15}$N) | - | 64 | - | - | 1h51m |
| TROSY-MUSIC[c](D+1, D) | 106.5($^1$H) | 16.4($^{15}$N) | - | 64 | - | - | 1h49m |
| TROSY-MUSIC[c](N+1, Q+1) | 106.5($^1$H) | 16.4($^{15}$N) | - | 128 | - | - | 3h47m |
| TROSY-MUSIC[c](N+1) | 106.5($^1$H) | 16.4($^{15}$N) | - | 128 | - | - | 3h45m |

[a] Experiments were performed at two temperatures: 298 K and 308 K.

[b] Experiments were performed with samples in D2O.

[c] Parameters and pulse sequences of the TROSY type MUSIC with semi constant time in indirect dimension were previously described in [39].

chemical shifts, TALOS-N uses a database of sequences to predict the secondary structure [41].

The secondary X-ray-based structures were extracted with the UCSF Chimera program [42] using the PDB entry: 7K7P [30].

In the text and figures, the standard nomenclature for amino acids of the carbon atoms was used, where $^{13}$Cα is the carbon next to the carbonyl group $^{13}$C′ and $^{13}$Cβ is the carbon next to $^{13}$Cα [43].

## Result and discussion

### Assignment protocol

The validity of the secondary structure analysis based on chemical shifts (CS) depends not only on the knowledge of CS of the $^1$HN, $^{15}$N, $^{13}$Cα, $^{13}$Cβ, $^{13}$C′ nuclei but also on $^1$H$^α$ assigned resonances. Thus, we used the fully protonated $^{15}$N,$^{13}$C-labelled SARS-CoV-2 nsp1. Additionally, this provides a foundation for validating inter β-strands interactions in the folded domain of the protein through observation of NOE contacts between $^1$HN-$^1$HN or H$^α$- $^1$HN protons

[44]. Recently, it was reported [31] that the secondary structure of the full-length SARS-CoV-2 nsp1 protein (Fig 1A) at pH 6.5 embraces one folded domain (residues 14–125) and two disordered chains, flanking the folded domain at the N-terminus (residues 1–13) and the C-terminus (residues 126–180).

It is known that folded and intrinsically disordered proteins (IDP) entities, have differences in relaxation properties and, thus, require different optimal NMR experimental conditions. This also means that the positions of the amide $^1$H and $^{15}$N chemical shifts of the disordered fragments and some flexible parts of the folded domains strongly depend on the buffer conditions including pH and temperature. Lower temperatures generally favor IDPs by reducing amide proton exchange while higher temperatures favor folded proteins due to reduced T2 relaxation. Keeping this in mind, we performed all NMR data collections at two temperature, 298 K and 308 K. This was proved to be sufficient for generating an almost complete assignment of the protein. Thus, we achieved the maximum NMR performances for the folded domain at 308 K and for the disordered fragments at 298 K.

We also performed all assignments at physiological buffer (pH 7.5), and low salt concentration, and in the absence or presence of 2.5 mM MgCl$_2$. These conditions are optimal for future analysis of nsp1 interaction with 40S ribosome subunit or RNA, where the presence of Mg$^{2+}$ could be expected. No differences in the spectra were observed in the presence or absence of MgCl$_2$. All data presented are from the spectra with MgCl$_2$.

As there are no programs readily available for performing automatic assignment of a protein containing both folded and disordered regions, we use the conventional manual assignment strategy based on experiments presented in Table 1. To achieve the best resolution in the 3D experiments in the indirect dimensions and to resolve resonances corresponding to the disordered part of protein, NMR experiments were mostly performed with the NUS option [33].

To facilitate the assignment procedure in this study, we chose the following strategy. First, amino acid selective TROSY- MUSIC experiments on full length wild type $^{15}$N,$^{13}$C labelled SARS-CoV-2 nsp1 at 308 K were performed. Fig 1 presents the superposition of the $^1$H-$^{15}$N HSQC of SARS-CoV-2 nsp1 protein (shown in grey) with TROSY-MUSIC spectra of selected D + 1, D, E + 1, A + 1, A, S + 1 and S cross peaks of amino acids. As we have described earlier [39], these type of experiments mostly benefits the analysis of IDPs.

Indeed, the TROSY-MUSIC spectra of SARS-CoV-2 nsp1 presented in (Fig 1A and 1B) show all D + 1, D, E + 1, E cross peaks from the disordered fragments, but not from the folded domain, and thus, allow easy assignment. However, TROSY-MUSIC with A + 1, A, S + 1 selection showed almost all expected correlations for their respective types of amino acid throughout the entire sequence of the nsp1 protein. The exceptions were the amide protons involved in slow conformational exchange, whose cross peaks in the $^1$H-$^{15}$N HSQC spectrum were broadened below the detection limit.

These data were used to assign the resonances at 308 K. The $^1$H-$^{15}$N-HSQC spectrum at 308K shows well-dispersed and narrow-line widths of the amide signals (Fig 2B and 2C). At this temperature, we have observed and assigned 158 residues, including prolines. Importantly, even at this higher temperature (308 K), amino acids 125K, 124R, 123L and 122L show two sets of amide $^1$HN - $^{15}$N cross peaks, which allowed us to conclude that the amino acids between the folded domain and the C-terminal disordered part of SARS-CoV-2 nsp1 protein adopt two distinguishable conformations detectable in the NMR time scale.

Next, we examined the broadening of some of the NH backbone resonances below the detection limit at 308 K. To perform this analysis and additionally validate resonances, which were ambiguously assigned due to the crowdedness of the spectra, we used NMR data of two SARS-CoV-2 nsp1 protein mutants. The selection of mutants was based on the following criteria: (1) the replacement of chosen amino acid should not lead to any strong conformational

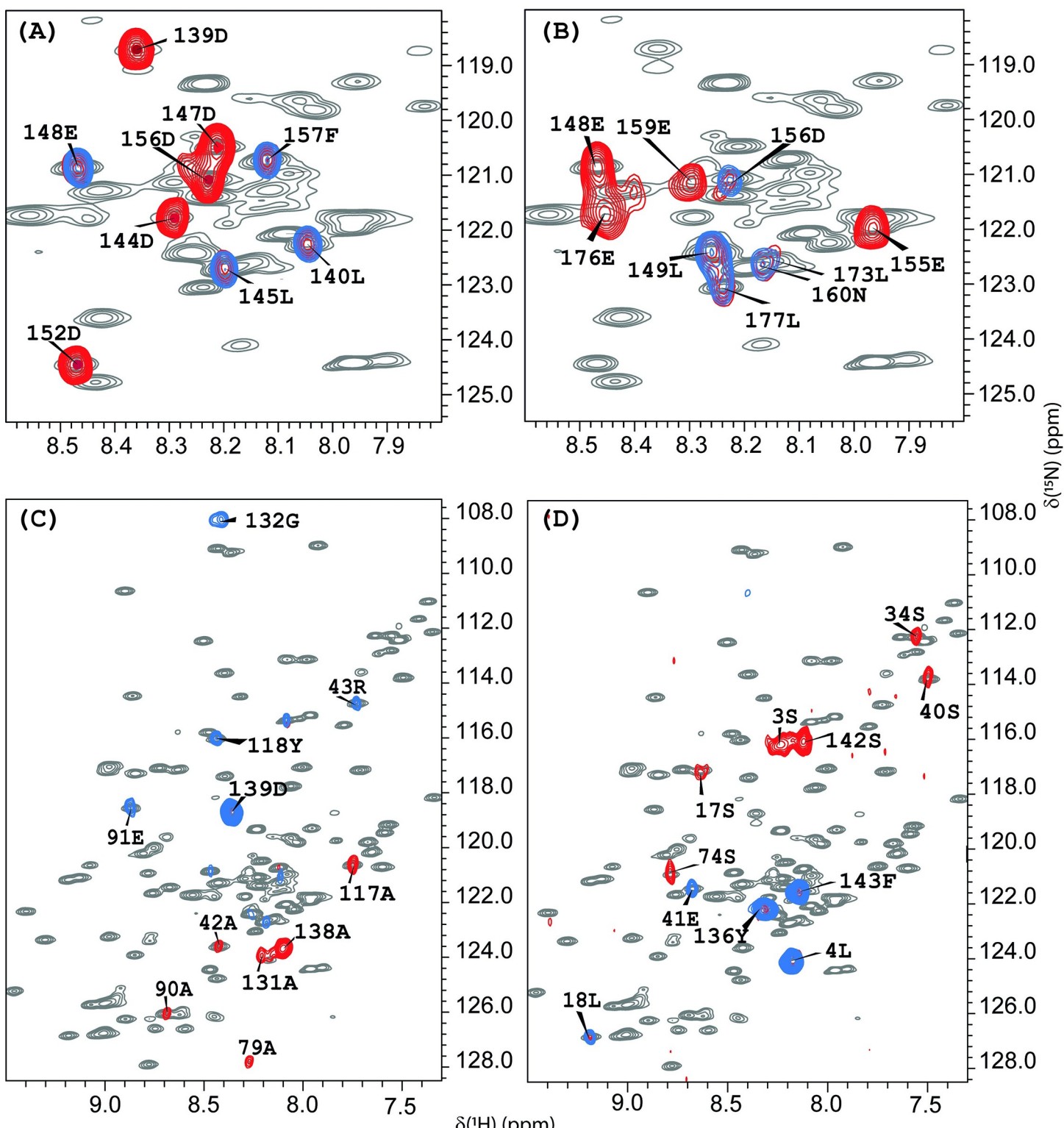

**Fig 1. $^{1}$H-$^{15}$N HSQC spectra at T = 308K of the full-length SARS-CoV-2 nsp1 protein with TROSY MUSIC experiments.** Superpositions of the $^{1}$H-$^{15}$N HSQC spectra of the full-length SARS-CoV-2 nsp1 protein (grey) (at 308 K) with TROSY MUSIC experiments. (**A**) Blue colour indicates D + 1, and red indicates D and D+1 cross peaks; (**B**) blue colour indicates E + 1, and red shows E and E + 1cross peaks; (**C**) blue colour shows A + 1, and red indicates A and A +1 cross peaks; (**D**) blue colour shows S + 1, and red indicates S and S + 1 cross peaks. The assignment of the observed cross peaks was done according to the amino acid sequence shown in Fig 2A.

**(A)**

¹MESLVPGFNE KTHVQLSLPV LQVRDVLVRG FGDSVEEVLS EARQHLKDGT CGLVEVEKGV
LPQLEQPYVF IKRSDARTAP HGHVMVELVA ELEGIQYGRS GETLGVLVPH VGEIPVAYRK
VLLRKNGNKG AGGHSYGADL KSFDLGDELG TDPYEDFQFN WNTKHSSGVT RELMRELNGG¹⁸⁰

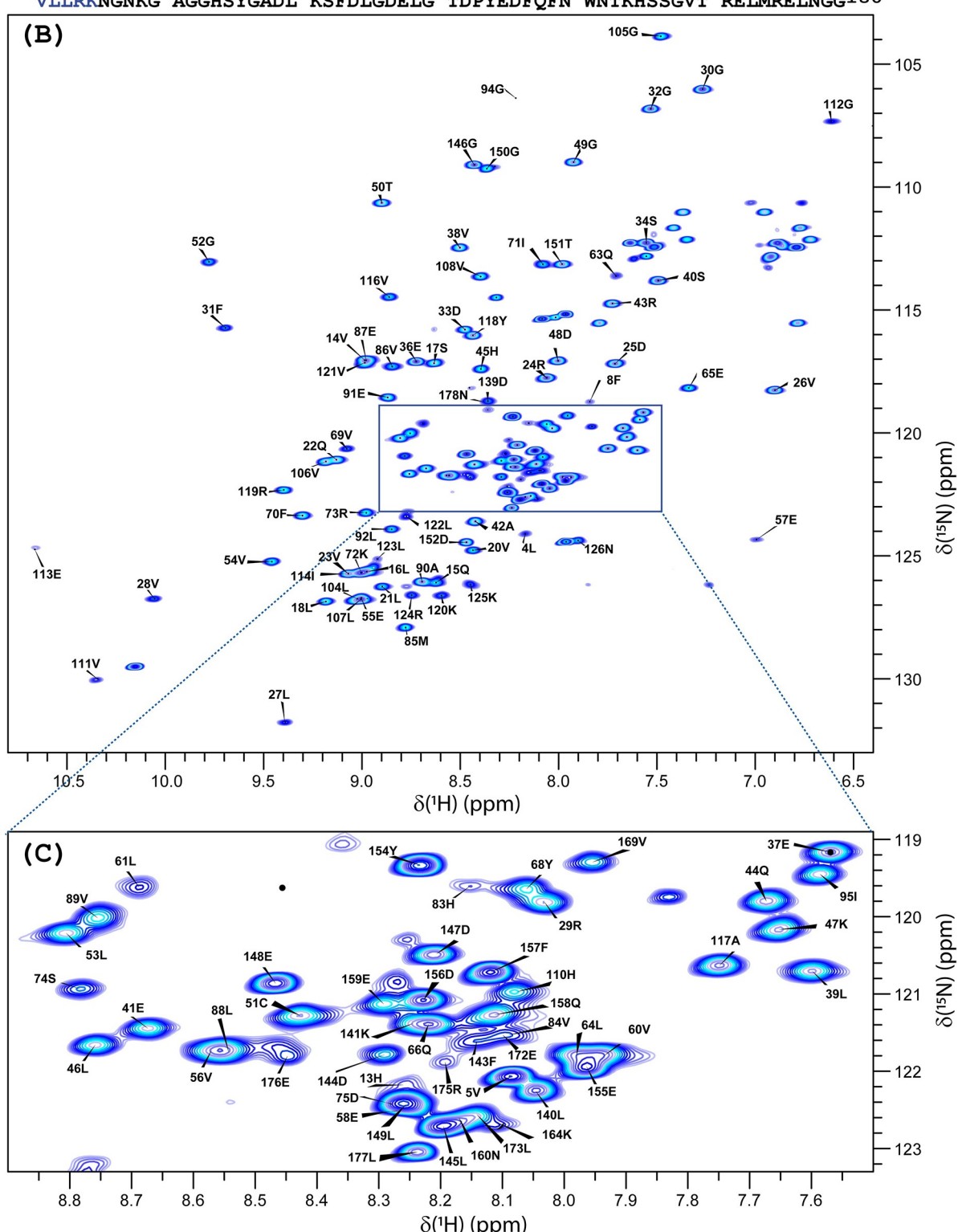

**Fig 2. $^1$H-$^{15}$N HSQC spectra of the full-length SARS-CoV-2 nsp1 protein at T = 308K with backbone amide assignments.** (**A**) Amino acid sequence of the full-length SARS-CoV-2 nsp1 protein. The folded domain is indicated in blue. (**B**) $^1$H-$^{15}$N HSQC spectrum with its extended crowded part (**C**) of the full-length $^{15}$N, $^{13}$C-labelled SARS-CoV-2 nsp1 protein in the buffer containing 20 mM HEPES pH 7.5, 100 mM KCl, 2.5 mM MgCl$_2$, 1 mM TCEP, 1 mM NaN$_3$, 10 (v/v) % D$_2$O and 0.1 mM DSS. The chemical shift assignment of NH backbone is shown by the number and letters corresponding to the amino acid sequence shown in panel (**A**). The assignment is presented only for the residues whose cross peaks were observed at 308 K.

transition in the protein and (2) preferably large chemical shift perturbation (CSP) should be expected in the place of the substituted amino acid. We have analysed NMR data of 2 mutants. In the first, a single histidine in position 81 was replaced by proline (H81P). According to the X-ray structure (PDB: 7K7P), the aromatic ring of H81 is located in a flexible loop and turned towards the solvent. Thus, its replacement should not lead to any significant change in the architecture of the secondary structure of SARS-CoV-2 nsp1 protein. Moreover, the replacement by proline was expected to induce a large perturbation of chemical shifts (CS) of the nuclei located in its proximity due to its unique structure and possibly enhance the stability of the loop. As it is shown in Fig 3D, the CSP observed in $^1$H-$^{15}$N HSQC spectra of wild type nsp1 vs the H81P mutant (red bars) are evident. As expected, the most significant CSPs were observed between residues 75 and 85. Noteworthy, the amino acids of the N-termini (10–17) and at the beginning of the C-terminal disordered fragment (residues 120–127) are affected as well. This finding led us to the conclusion that amino acids corresponding to those three regions are in close proximity.

In the second mutant, we introduced two aa substitutions: Lys129 and Asp48 were replaced by Gln (K129E and D48E). The first mutation is in the beginning of the disordered fragment of SARS-CoV-2 nsp1. The second mutation, according to the X-ray structure (PDB: 7K7P), is located at the end of the α-helix. In Fig 3D, the CSPs observed in $^1$H-$^{15}$N HSQC spectra of wild type nsp1 vs the double mutant are presented by blue bars. Significant CSPs were observed for residues 44–51 and 125–132.

The strategy of inducing chemical shift changes in particular regions of the protein by carefully selected replacements of structurally insensitive amino acids was very valuable and helped us to resolve some ambiguities in the backbone and side chain assignments. More detailed characterisation of the SARS-CoV-2 nsp1 mutants will be published elsewhere.

The resulting assignment of the full length of SARS-CoV-2 nsp1 was as following. For the folded domain (residues 14–125) we assigned 95% $^1$HN and 95% $^{15}$N including prolines, 96% of $^{13}$Cα, 88% of $^{13}$Cβ, 94% of $^{13}$C′ and 83% of all H$^\alpha$. For the two disordered fragments (residues 1–13 and 126–180) we assigned 91% $^1$HN, 91% $^{15}$N, 94% of $^{13}$Cα, 93% of $^{13}$Cβ, 94% of $^{13}$C′ and 76% of all H$^\alpha$.

Comparison of NMR spectra acquired in different conditions in this and published [31] work revealed only small changes for folded, dynamically stable nsp1 domain. In that study, the authors have assigned more of the amide resonances, and this is not surprising considering their lower pH experimental conditions. However, in our assignment, there are some residues that were previously missed: Q63, E102 and T103. Only one residue shows significant difference: N126. The resonances corresponding to the loops and the disordered regions of the protein were more likely to be affected by changes in temperature and pH, but they do not indicate any profound structural difference. Residues showing chemical shift differences larger than 0.2 are: A42, H45, C51, V111, N126, S135 and S166. Differences between 0.1 and 0.2 were observed for: D48, G49, G52, V56, E57, V60, L61, R77, H83, V108, H110, E113 and I114. The differences were calculated using: $\sqrt[2]{\left(\Delta^1H\right)^2 + \left(0.15\Delta^{15}N\right)^2}$. All $^1$H, $^{15}$N and $^{13}$C chemical shifts of the full-length of SARS-CoV-2 nsp1 protein at pH 7.5 and at two temperatures, 298 K

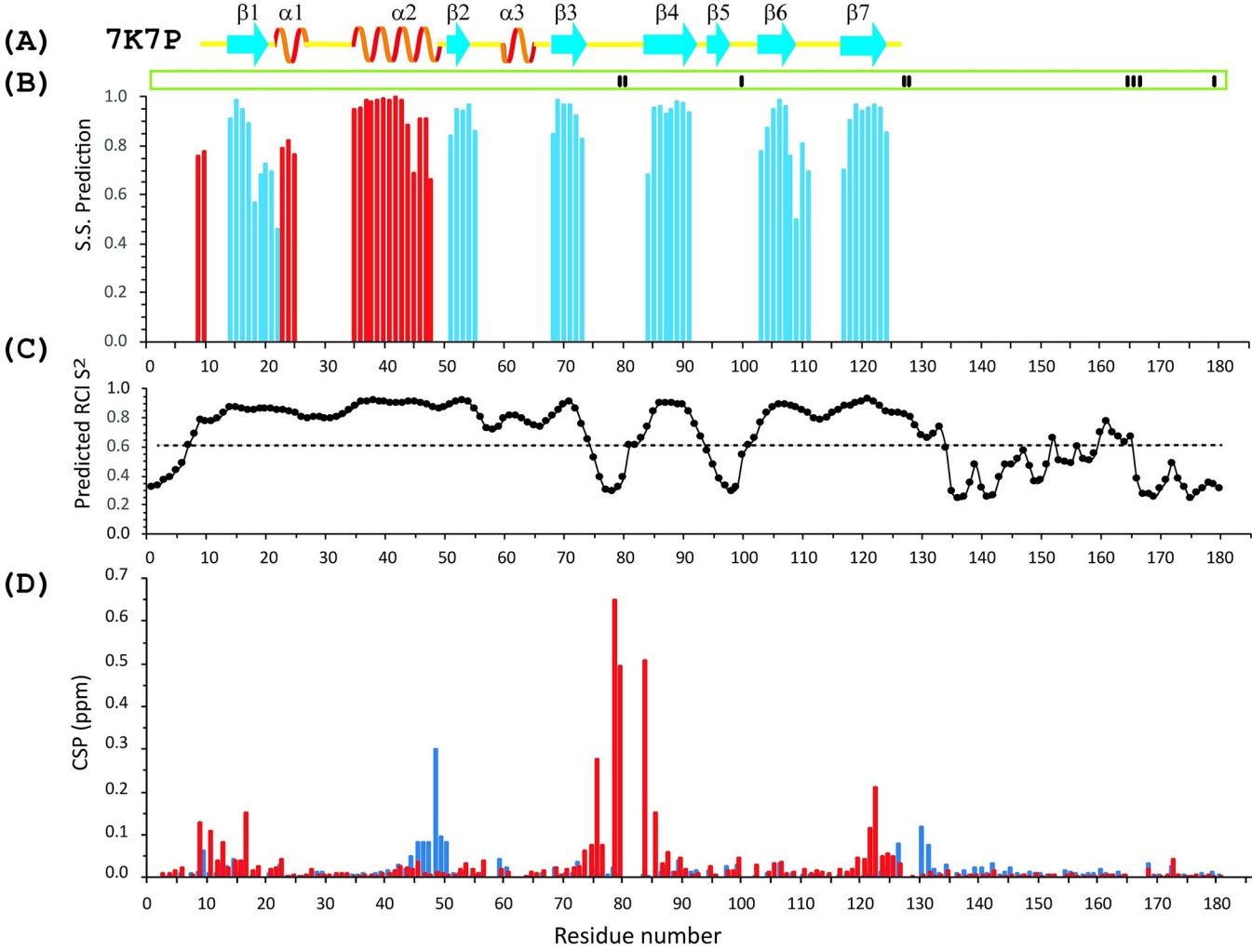

**Fig 3. Secondary structure and order parameters of the full-length SARS-CoV-2 nsp1 protein.** (**A**) The secondary structure derived from the X-ray data of the folded domain of SARS-CoV-2 nsp1 (residues 10–124, PDB: 7K7P) according to UCSF Chimera [42]. The yellow lines show loop segments of the protein. (**B**) Index of secondary structure prediction (S.S. Prediction) (*red* and *blue* bars indicate α-helices and β-strands, respectively. (**C**) Model-free [45] order parameter, S$^2$, [46] extracted by TALOS-N [41]. (**D**) Chemical shift deviations of Δ$^1$H and Δ$^{15}$N nuclei (ppm), obtained as distances $\sqrt[2]{(\Delta^1\text{H})^2 + (0.15\Delta^{15}\text{N})^2}$, between wild type SARS-CoV-2 nsp1 protein and nsp1(H81P) (red) or nsp1(K129E, D48E) (blue) mutants. Amino acids which did not contribute to the TALOS-N analysis are shown by short black bars in the green box on top of panel (**B**).

and 308 K, have been deposited in BioMagResBank (http://www.bmrb.wisc.edu) under the accession 50915. The chemical shifts for the earlier published work have been taken from Bio-MagResBank accession 50620 (31).

## Secondary structure of the SARS-CoV-2 nsp1 protein

The full-length SARS-CoV-2 nsp1 protein chemical shifts were analyzed with TALOS-N, and the data are presented in Fig 3B and 3C. The chemical shifts of the main peaks were used in the cases where dual peaks were observed. The analysis of the secondary structures of the folded domain of the SARS-CoV-2 nsp1, which was derived from the NMR data, and the previously determined crystal structure (7K7P) shows that they are almost identical. This additionally validated our assignment (Fig 3A and 3B). The secondary structure of the earlier

assigned SARS-CoV-2 nsp1 and that determined in our study are also similar. Nevertheless, a few important inconsistencies are evident.

The crystal structure of SARS-CoV-2 nsp1 folded domain [28–30] revealed the presence of an additional short β5-strand (residues 95–97), which is not found in the structure of SARS-CoV nsp1 determined by NMR (PDB: 2HSX). In our study of the full-length nsp1, the presence of β5-strand folded between residues 95 and 97 could not be confirmed, neither was it found in the NMR assignment by Wang et al (31). Moreover, according to the predicted order parameter [$S^2$ by TALOS-N ([Fig 3C])], the segment between residues 92–103 is dynamic. This prediction is in agreement with our finding that amide protons between residues I95 and G98 were not observed at both 308 K and 298 K, suggesting their involvement in multiple conformational exchange and exposure to the solvent. We additionally performed the analysis of the 3D NOESY $^{15}$N -HSQC spectrum to determine dipole-dipole contacts of NH-NH and NH-H$^\alpha$ protons, which allows the detection of hydrogen bonds between two β-strands [44]. The β-sheet formed by strands β4 and β3, according to the X-ray structure, was confirmed by observing NH-NH and NH-Hα NOEs between those strands, but not between the β4 and β5 strands. These data contradict the X-ray results, which suggest low mobility of the β5 strand due to the additional hydrogen bonds between the β4 and β5 strands.

Subtle differences between the X-ray and NMR secondary structures were also noticed for the strands β1, β2, β6 and β7. According to our NMR data, in solution, these strands are extended by one or two amino acids at their C-termini. Furthermore, the α-helix 2 in solution is one amino acid shorter than in the X-ray structure. Our data also predicted that the α-helix has a break at residue H45. The X-ray structure also shows two short 3–10 helices between amino acids 22 to 26 and 60 to 64, respectively (helix 1 and 3). The TALOS-N analysis of our chemical shifts detected the first helix, albeit shorter (residues 23 to 25), but not the second one. One should note that TALOS-N does not differentiate between α and 3–10 helices but classifies both types as helices [41]. A perusal of possible NOE cross peaks of the two regions shows some indication of helix formation in the first region, but none in the second. In conclusion, we observed a short helix corresponding to α1, but did not observe α3. This is the same result as in the previous assignment of nsp1 at pH 6.5 published by Wang et al (31). Instead, our chemical shifts suggest the presence of a long, disordered loop between residues 55 and 67, which, according to the TALOS-N prediction, has restricted mobility ([Fig 3B and 3C]). Importantly, this region of the SARS-CoV-2 nsp1 sequence was well characterised by NMR through chemical shifts as well as by NOE of NH-NH and NH-Hα proton contacts. This led us to conclude that these discrepancies between NMR and X-ray secondary structure predictions likely result from the crystallisation conditions. Interestingly, Wang et al suggests the possibility of a short helix consisting of residues 171–175 albeit with lower probability than those of other helices in the protein (31). Our data does not support this. This could be an effect of the differences in pH between our studies.

Two more dynamic regions in the solution structure of the folded domain of SARS-CoV-2 were identified based on the $S^2$ order parameter predicted by TALOS-N: S74-H83 and L92-E102 ([Fig 3B and 3C]). This prediction is in line with the lack of peaks or broadened $^{15}$N/$^1$HN cross peaks, even at 298 K in the $^1$H-$^{15}$N NMR spectra. We did not observe resonances for H81 and G82 in the first region nor for S100 and G101 in the second one. It can be explained either by broadening of the HSQC cross peaks below detection limit or, more likely, by the involvement of these regions in slow conformational exchange. This was observed in the earlier NMR assignment of nsp1 as well. In that study resonances of E93, Q96, E102 and T103 could not be assigned (31).

The N- and C-termini, comprising amino acids M1-N9 and N124-G180, respectively, were identified by CSI as fully unstructured, but showing differences in the predicted order

parameters ($S^2$) throughout the sequences. Dynamic regions with an order parameter $S^2$ below 0.6 were predicted for M1-F8, S135-Q158 and H165-G180. The increase in dynamic behaviour of those residues correlated with the changes in the intensities of the amide backbone cross peaks in the $^1$H-$^{15}$N HSQC spectra of nsp1. These cross peaks have higher intensity compared to cross peaks belonging to the amino acids in folded, less dynamic regions. Other residues in the unstructured regions are less dynamic.

For the H81P mutant, we see chemical shift changes for residues 10–17 and 120–127 belonging to the junctions between structured/unstructured regions (Fig 3D). Based on these data, we propose that in the full-length SARS-CoV-2 nsp1, the folded and disordered parts of the protein behave not as fully independent units but are rather involved in intramolecular interactions. This has also been suggested by Wang et al, who hypothesised that the unstructured C-terminal region interacts with the folded region (31). This may stabilize the overall protein fold and improve its solubility. The second set of chemical shifts of the residues having a double set of signals, amino acids 122–125, was also analysed by TALOS-N combined with the other chemical shifts of residues with single cross peaks, but no significant change from the main set could be observed. This was expected as the chemical shift differences are small between the sets. Comparing the calculated $S^2$ values of this region between our data and those obtained by Wang et al reveals some differences. In our study, the $S^2$ stays above 0.75 for all residues involved but, in their case, the $S^2$ drops below 0.75 for the last two residues and continues to drop in the following residues to levels typical for unstructured domains (31). In our case, the $S^2$ drops to the level of unstructured domains only after G133. As mentioned above, we suggest that residues 120–127 may be involved in interaction(s) with the folded domain, probably with the dynamic loop between residues 74–83. It is reasonable to speculate that this difference between the two NMR assignments is the result of using different pH and salt concentration. It is tempting to further speculate that this interaction may result in multiple protein conformations involving the C-terminal unstructured domain and that this might explain the multiple functions of nsp1 in viral replication and virus-host interactions. It is in our plans to further investigate this phenomenon possibly by some mutations in the 120–127 region.

In conclusion, the near complete $^{15}$N/$^{13}$C/$^1$H backbone resonance and part of side chain assignment of the full-length SARS-CoV-2 nsp1 at pH 7.5 and physiological salt concentration has been performed. Validation of assignment have been done by using two different nsp1 mutants as well as MUSIC type amino acid selective experiments. Assignment revealed that the secondary structure of the rigid folded domain is almost identical to that determined by X-ray. However, the existence of the short β-strand (residues 95 to 97), which is considered to be the significant structural difference between SARS-CoV-1 and SARS-CoV-2 nsp1 proteins, has not been confirmed. In solution, SARS-CoV-2 nsp1exhibits disordered, flexible N- and C-termini, having different dynamics. The short peptide in the beginning of the C-terminal disordered fragment (122–125) adopts two conformations. We propose that there are intramolecular interactions between the disordered and folded nsp1 domains, most likely involving the region two conformations. Studies of the structure and dynamics of the SARS-CoV-2 mutants in solution are on-going and will provide important insights on the molecular bases underlying these interactions. Further mutations and interactions with RNA and other proteins are under way as well.

## Acknowledgments

We thank Nikita Shiliaev for technical assistance.

## Author Contributions

**Conceptualization:** Ilya Frolov, Elena I. Frolova.

**Data curation:** Tatiana Agback, Ilya Frolov, Peter Agback.

**Formal analysis:** Tatiana Agback, Peter Agback.

**Funding acquisition:** Ilya Frolov, Peter Agback.

**Investigation:** Tatiana Agback, Francisco Dominguez, Ilya Frolov, Elena I. Frolova, Peter Agback.

**Methodology:** Tatiana Agback, Francisco Dominguez.

**Project administration:** Ilya Frolov, Peter Agback.

**Resources:** Elena I. Frolova.

**Validation:** Tatiana Agback.

**Visualization:** Tatiana Agback.

**Writing – original draft:** Tatiana Agback, Ilya Frolov, Elena I. Frolova.

**Writing – review & editing:** Tatiana Agback, Elena I. Frolova, Peter Agback.

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
