## [Decision Letter · Decision Letter 0]

10 Jun 2021

PONE-D-21-14234

1H, 13C and 15N resonance assignment of the SARS-CoV-2 full-length nsp1 protein and its mutants reveals its unique secondary structure features in solution.

PLOS ONE

Dear Dr. Agback,

Thank you for submitting your manuscript to PLOS ONE. After careful consideration, we feel that it has merit but does not fully meet PLOS ONE’s publication criteria as it currently stands. Therefore, we invite you to submit a revised version of the manuscript that addresses the points raised during the review process.

Please be aware that both reviewers raised a significant number of concerns that should be completely addressed in a revised version of the manuscript, which will likely undergo a second round of review. Of particular importance is to clarify the novelty of the work, in relation to previous publications.

We look forward to receiving your revised manuscript.

Kind regards,

Oscar Millet

Academic Editor

PLOS ONE

Journal Requirements:

3. We note that Figure 1 and Figure 2 in your submission contain copyrighted images. All PLOS content is published under the Creative Commons Attribution License (CC BY 4.0), which means that the manuscript, images, and Supporting Information files will be freely available online, and any third party is permitted to access, download, copy, distribute, and use these materials in any way, even commercially, with proper attribution. For more information, see our copyright guidelines: http://journals.plos.org/plosone/s/licenses-and-copyright.

a. You may seek permission from the original copyright holder of Figure 1 and Figure 2 to publish the content specifically under the CC BY 4.0 license.

Additional Editor Comments (if provided):

Reviewers' comments:

Reviewer's Responses to Questions

**Comments to the Author**

1. Is the manuscript technically sound, and do the data support the conclusions?

Reviewer #1: Yes

Reviewer #2: Yes

2. Has the statistical analysis been performed appropriately and rigorously? 

Reviewer #1: Yes

Reviewer #2: N/A

3. Have the authors made all data underlying the findings in their manuscript fully available?

Reviewer #1: Yes

Reviewer #2: Yes

4. Is the manuscript presented in an intelligible fashion and written in standard English?

Reviewer #1: Yes

Reviewer #2: No

5. Review Comments to the Author

Reviewer #1: The manuscript describes the NMR chemical shfit assignment of the nsp1 protein from SARS-CoV-2. Unfortuneatly, it is the second report of this NMR resonance assignment. The authors rightly reference the previous work, but the differences e.g. construct design is not being discussed. The authors do not compare their chemical shift assignment with the previously published data.

The authors here also investigate mutants of nsp1, but the differences e.g in the dynamics of "wildtype" and mutant protein are not presented.

At this point, I cannot recommend publication of this work.

Reviewer #2: The submitted manuscript presents a far-reaching NMR analysis of the SARS-CoV-2 Nsp-1 protein, using both wild-type and two mutant proteins, and at two temperatures. For this, the authors employ a very complete and high-level set of NMR experiments, going beyond the standard protocol and addressing (by MUSIC type experiments) the specific needs for proteins with large unstructured regions (as in Nsp-1). Thus, the work deserves publication for its sound technical implementation and for the high relevance of the targeted protein. However, the manuscript requires a minor-to-mayor overhaul for the reasons listed below, where especially one critical prior work (correctly cited as ref. 31) must be considered more thoroughly. Of note, while this prior work formally limits the novelty of the presented manuscript, this does not constitute a valid reason for rejection in my eyes - rather, I "merely" request a more detailed comparison (stressing the agreements and working out the minor differences) to consolidate, deepen, and extend our understanding of this functionally important protein.

General requests:

- I recommend that a native English speaker revises the manuscript as it contains awkward wording in several larger parts of the text (e.g., legend of Fig. 1, text paragraphs l. 211-217, l. 269-272, 320-324, 334-336) that strongly compromises readability.

- The cited reference [31] reports critical prior work and must be considered in more detail. Thus, the authors should work out more clearly, and wherever relevant in the manuscript, the main differences and agreements between their new work and ref. [31]. For instance: For which further residues were the authors able to provide a NMR characterisation? Why do the authors find slightly different residue segments for the unstructured N- and C-terminal and for the structure central regions? Why could the authors not confirm the short C-terminal helix H3? Could such differences be due to their use of an older version of TALOS (TALOS+ instead of TALOS-N), or to the slightly different experimental conditions (pH, temperature, etc.)?

Specific requests:

- Table 1 (NMR parameters): Since recorded FID resolution is the inverse of the maximal sampling time (i.e., FID resolution = 1/AQ, which should be stated clearly in the caption to column 2), very low F1 and F2 resolutions follow from the listed data and I wonder whether there might be a systematic error (factor 2?) in these numbers? For instance, the listed 12.3 ms for 15N result in only 81 Hz = 1.35 ppm resolution. Such low resolution makes little sense especially for intrinsically disordered proteins/regions (as present in Nsp-1). Moreover, it invalidates both resolution enhancing NMR methods adopted by the autors, i.e. non-uniform sampling (NUS) and semi-constant t1(15N) evolution. Already constant-time evolution of 1J(NCO) = 15 Hz or 1J(NCA) = 9-12 Hz demands transfer delays at least twice as long as the cited 12.3 ms (i.e., up to 30 ms), and semi-constant time evolution would require even longer delays. Please clarify this obvious contradiction. I also suggest to add a further column to Table 1 listing the pertaining total measurement time for each experiment.

- L. 134: state that 14.1 T correspond to 600 MHz for 1H to enable conversion of resolution, from Hz to ppm (as required to convert the maximal evolution times into recorded FID resolutions, see my notes on Table 1 above).

- L. 175/176: It is stated that folded and disordered regions show distinct NMR relaxation properties exacting distinct optimal “NMR experimental conditions”. This should be better elaborated. For instance: lower temperatures favour NMR on IDP due to reduced HN/H2O exchange while higher T favour folded regions due to reduced T2 relaxation. Mention further NMR parameters like transfer and interscan delays that also differ – how was this implemented or considered?

- Terminology: “DIPSI2-N-HSQC” and “MLEV17-C-HSQC” (line 141) are unconventional names (taken directly from the BRUKER pulse program library?) – better use the standard NMR experiment name = TOCSY-HSQC (or was it a HSQC-TOCSY?). Also, in line 143, I suppose the authors meant to say “NOESY-HSQC” (or HSQC-NOESY)?

- L.143: HA assignment should, in principle, follow directly from a combination of HACACO with HNCO and HN(CO)CA spectra, making the acquisition of both NOESY-HSQC meaningless and superfluous for this purpose. I suppose, however, that these latter spectra were rather recorded with the objective to search for structure indication? If so, please indicate!

- The optional addition of MgCl2 (2.5 mM) is unclear to me and should be explained. How did it affect the NMR spectra?

- The abbreviation “aa” should be avoided and spelled out: “amino acid”, or “residue”

- The short helix a3 appears to be missing in the NMR structure, but shows indication of restricted mobility (lines 309 – 312). As a matter of fact, the X-ray structure rather reports a short 3-10 (instead of canonical alpha) helix here. Thus, have the authors considered the possible presence of a 3-10 helix? What secondary structure does the CSI for the HA nuclei suggest (considering that helicity indicated only by HA may suggest a 3-10 helix)?

- Terminology: While technically correct, the term “cross peak” for an HSQC signal appears very confusing as it is conventionally reserved to NOE cross peaks. This may give rise to confusions, for instance, when reading through lines 328-330 where the disordered regions of Nsp-1 are discussed, but where no (NOE) cross signals are expected.

- This confusion is accentuated by the subsequent conclusion of the authors (lines 330-332, and section “Conclusion”) that folded and unfolded regions in Nsp-1 would show some kind of intramolecular interactions. As it stands, this conclusion is not substantiated enough (although the prior work – ref. 31 – also suggested this from the notable solubility enhancement of the C-terminus on the folded domain).

- I was intrigued by the finding of a double set of signals for residues 122-125 (line 221). Then, however, I wonder which of both signal shifts (for each residue) was used for the secondary structure analysis shown in Fig. 3b, and how the predicted secondary structuredness differs for both sets of shifts? This should be clarified and the alternative signal shifts should be reported more clearly – ideally, by inclusion into Fig. 3b (e.g., as a second set of coloured bars). Apparently, there is someconformational dynamics towards the C-terminus of strand b7: could this possibly be related to a “kink” in this strand, near V121? The competing work in ref.[ 31] also mentions such conformational heterogeneity (without further specifying the residues) and suggests a possible role of cis/trans isomerism at a proline residue (again, without specifying the residue number, but possibly referring to P115). Interestingly also, the RCI-S2 order parameters derived by the authors and in ref. [31] differ more strongly in exactly this region (res. 120 – 130), where the authors derive a significantly higher rigidity than reported in [31]. I would greatly welcome if the authors provided more details and some deeper discussion of this interesting observation of conformational exchange exactly in the transition region between folded and unfolded (C-terminal) regions, as this might even have a relevance for the known “switch” in Nsp-1 function (from host translation inhibition to viral translation initiation).

6. PLOS authors have the option to publish the peer review history of their article (what does this mean?). If published, this will include your full peer review and any attached files.

Reviewer #1: No

Reviewer #2: **Yes: **Tammo Diercks

---

## [Author Response · Author response to Decision Letter 0]

17 Sep 2021

Here are our answers to the concerns and questions of the reviewers.

Reviewer #1: The manuscript describes the NMR chemical shfit assignment of the nsp1 protein from SARS-CoV-2. Unfortuneatly, it is the second report of this NMR resonance assignment. The authors rightly reference the previous work, but the differences e.g. construct design is not being discussed. 

The authors do not compare their chemical shift assignment with the previously published data.

The authors here also investigate mutants of nsp1, but the differences e.g in the dynamics of "wildtype" and mutant protein are not presented.

At this point, I cannot recommend publication of this work.

Answer: As an explanation to the missing comparison, the other assignment paper came out just as our MS was finished. Despite the publication, the chemical shifts were not available at that time and thus, we only added a few sentences about this. In the revised version, since now we have the data, we have heavily modified the MS with considerable comparison with the previous work and more discussion about our results. We hope you will look on the new draft more positively. Note that the mutations were only done to help in the assignment and have no biological meaning. Our biological partners are considering to test them for any effect and if so, we will return to them with a further study.

Reviewer #2: The submitted manuscript presents a far-reaching NMR analysis of the SARS-CoV-2 Nsp-1 protein, using both wild-type and two mutant proteins, and at two temperatures. For this, the authors employ a very complete and high-level set of NMR experiments, going beyond the standard protocol and addressing (by MUSIC type experiments) the specific needs for proteins with large unstructured regions (as in Nsp-1). Thus, the work deserves publication for its sound technical implementation and for the high relevance of the targeted protein. However, the manuscript requires a minor-to-mayor overhaul for the reasons listed below, where especially one critical prior work (correctly cited as ref. 31) must be considered more thoroughly. Of note, while this prior work formally limits the novelty of the presented manuscript, this does not constitute a valid reason for rejection in my eyes - rather, I "merely" request a more detailed comparison (stressing the agreements and working out the minor differences) to consolidate, deepen, and extend our understanding of this functionally important protein.

Answer: Thank you for taking time to go our manuscript in great detail. When ref 31 came out we had finished our MS and as at that time, we did not yet have access to their chemical shifts, we could not do a proper comparison. Now with the data availability, we have added more text to the MS that discusses differences as well as similarities with the earlier study. We hope that you will find it satisfactory.

General requests:

- I recommend that a native English speaker revises the manuscript as it contains awkward wording in several larger parts of the text (e.g., legend of Fig. 1, text paragraphs l. 211-217, l. 269-272, 320-324, 334-336) that strongly compromises readability.

Answer: We have gone through the parts indicated again and had a native English speaker helping us. We hope this have increased the understanding of the text.

- The cited reference [31] reports critical prior work and must be considered in more detail. Thus, the authors should work out more clearly, and wherever relevant in the manuscript, the main differences and agreements between their new work and ref. [31]. For instance: For which further residues were the authors able to provide a NMR characterisation? Why do the authors find slightly different residue segments for the unstructured N- and C-terminal and for the structure central regions? Why could the authors not confirm the short C-terminal helix H3? Could such differences be due to their use of an older version of TALOS (TALOS+ instead of TALOS-N), or to the slightly different experimental conditions (pH, temperature, etc.)?

Answer: As we are doing a major revision, we decided to upgrade our analysis to TALOS-N. No major changes were observed. We have added discussion and comparison to ref 31 throughout the manuscript.

Specific requests:

- Table 1 (NMR parameters): Since recorded FID resolution is the inverse of the maximal sampling time (i.e., FID resolution = 1/AQ, which should be stated clearly in the caption to column 2), very low F1 and F2 resolutions follow from the listed data and I wonder whether there might be a systematic error (factor 2?) in these numbers? For instance, the listed 12.3 ms for 15N result in only 81 Hz = 1.35 ppm resolution. Such low resolution makes little sense especially for intrinsically disordered proteins/regions (as present in Nsp-1). Moreover, it invalidates both resolution enhancing NMR methods adopted by the autors, i.e. non-uniform sampling (NUS) and semi-constant t1(15N) evolution. Already constant-time evolution of 1J(NCO) = 15 Hz or 1J(NCA) = 9-12 Hz demands transfer delays at least twice as long as the cited 12.3 ms (i.e., up to 30 ms), and semi-constant time evolution would require even longer delays. Please clarify this obvious contradiction. I also suggest to add a further column to Table 1 listing the pertaining total measurement time for each experiment.

Answer: The data in the table is taken directly from the Bruker acqupars, so the experimental conditions are easy to reproduce if any reader wants to. The resolutions are actually slightly better than what the Bruker standard parameter files suggests. We are also using linear prediction to improve the resolution. This we have added to the experimental section as clarification.

However, it is an important question as one always need to balance the resolution with the relaxation properties of the molecule being investigated. In addition, one need to consider the time available on the spectrometer. This protein is not deuterated, and due to this, we have to be careful of the relaxation properties to avoid collecting noise instead of signals. We also worried about line broadening due to the higher, physiological pH that our virology partners insisted upon. We have in recent years worked a lot on IDPs from other viral proteins such as nsP3 HVD of chikungunya and Venezuelan equine encephalitis viruses, and we have found that the parameters presented are quite optimal for the IDPs. At least in our hands, the use of MUSIC experiments is much more efficient, both regarding time as well as assignment, than increasing the resolution for “normal” 2D and 3D spectra. In earlier work, we have used TA (targeted acquisition) for optimal resolution/spectrometer time, but the results were almost always very similar to those we describe in the table.

Interestingly, we ran all our experiments on our own 600 MHz, and we didn’t see any need to go to higher fields for this protein. In our planned dynamic measurements, we will of course use higher fields as well.

Yes, we will add a column with measurement time.

- L. 134: state that 14.1 T correspond to 600 MHz for 1H to enable conversion of resolution, from Hz to ppm (as required to convert the maximal evolution times into recorded FID resolutions, see my notes on Table 1 above).

Answer: Added: “corresponding to 600 MHz” to sentence.

- L. 175/176: It is stated that folded and disordered regions show distinct NMR relaxation properties exacting distinct optimal “NMR experimental conditions”. This should be better elaborated. For instance: lower temperatures favour NMR on IDP due to reduced HN/H2O exchange while higher T favour folded regions due to reduced T2 relaxation. Mention further NMR parameters like transfer and interscan delays that also differ – how was this implemented or considered?

Answer: Section has been rewritten and is hopefully clearer. We started by testing different temperatures and as that was enough to obtain almost complete assignment, we did not try to optimize other parameters.

- Terminology: “DIPSI2-N-HSQC” and “MLEV17-C-HSQC” (line 141) are unconventional names (taken directly from the BRUKER pulse program library?) – better use the standard NMR experiment name = TOCSY-HSQC (or was it a HSQC-TOCSY?). Also, in line 143, I suppose the authors meant to say “NOESY-HSQC” (or HSQC-NOESY)?

Answer: Yes, our nomenclature was unclear, MS changed to TOCSY-HSQC and NOESY-HSQC.

- L.143: HA assignment should, in principle, follow directly from a combination of HACACO with HNCO and HN(CO)CA spectra, making the acquisition of both NOESY-HSQC meaningless and superfluous for this purpose. I suppose, however, that these latter spectra were rather recorded with the objective to search for structure indication? If so, please indicate!

Answer: We have found that it is more effective timewise to run a NOESY-HSQC in order to get the HA and ignore the HACACO, and this obviously generated more information as well. We added an explanation to the text.

- The optional addition of MgCl2 (2.5 mM) is unclear to me and should be explained. How did it affect the NMR spectra?

Answer: No difference observed. Observation added to the manuscript. This was in preparation for future possible interaction studies with RNA.

- The abbreviation “aa” should be avoided and spelled out: “amino acid”, or “residue”

Answer: Changed throughout the manuscript. We have tried to minimize the use of other abbreviations as well in the revised manuscript.

- The short helix a3 appears to be missing in the NMR structure but shows indication of restricted mobility (lines 309 – 312). As a matter of fact, the X-ray structure rather reports a short 3-10 (instead of canonical alpha) helix here. Thus, have the authors considered the possible presence of a 3-10 helix? What secondary structure does the CSI for the HA nuclei suggest (considering that helicity indicated only by HA may suggest a 3-10 helix)?

Answer: Thank you for the suggestion. We have now checked out the possibility of a 3-10 helix. No evidence for such a helix can be found in our NOE or the predicted backbone torsional angles or using HA only. A discussion of this have been added to the text. Note that for this helix, we only have HA for V60 and Q63. We are missing those of L61, L64 and P62. No secondary structure was observed in ref 31 as well. One should note that helix-1, which is also a 3-10 helix in the x-ray structure, was identified by Talos-N.

- Terminology: While technically correct, the term “cross peak” for an HSQC signal appears very confusing as it is conventionally reserved to NOE cross peaks. This may give rise to confusions, for instance, when reading through lines 328-330 where the disordered regions of Nsp-1 are discussed, but where no (NOE) cross signals are expected.

Answer: We have always used cross peak to denote the signals in multi-dimensional spectra of all kinds in all our previous publications. We will rewrite LL328-330 as well as other lines mentioning” cross peaks” to make it clearer what we exactly mean.

- This confusion is accentuated by the subsequent conclusion of the authors (lines 330-332, and section “Conclusion”) that folded and unfolded regions in Nsp-1 would show some kind of intramolecular interactions. As it stands, this conclusion is not substantiated enough (although the prior work – ref. 31 – also suggested this from the notable solubility enhancement of the C-terminus on the folded domain).

Answer: You are right, as it was written the conclusion could not be made. The lines have been rewritten and expanded to make our point understandable (i.e. mutation suggests interaction with the unstructured parts). Comparison to ref 31 has also been added.

- I was intrigued by the finding of a double set of signals for residues 122-125 (line 221). Then, however, I wonder which of both signal shifts (for each residue) was used for the secondary structure analysis shown in Fig. 3b, and how the predicted secondary structuredness differs for both sets of shifts? This should be clarified and the alternative signal shifts should be reported more clearly – ideally, by inclusion into Fig. 3b (e.g., as a second set of coloured bars). Apparently, there is some conformational dynamics towards the C-terminus of strand b7: could this possibly be related to a “kink” in this strand, near V121? The competing work in ref.[ 31] also mentions such conformational heterogeneity (without further specifying the residues) and suggests a possible role of cis/trans isomerism at a proline residue (again, without specifying the residue number, but possibly referring to P115). Interestingly also, the RCI-S2 order parameters derived by the authors and in ref. [31] differ more strongly in exactly this region (res. 120 – 130), where the authors derive a significantly higher rigidity than reported in [31]. I would greatly welcome if the authors provided more details and some deeper discussion of this interesting observation of conformational exchange exactly in the transition region between folded and unfolded (C-terminal) regions, as this might even have a relevance for the known “switch” in Nsp-1 function (from host translation inhibition to viral translation initiation).

Answer: Longer discussion about the double sets of signals included. Yes, we ran Talos with the second set, but hardly any difference was observed. The chemical shift differences were quite minor so there was no surprise there. This is the same region that we think have some kind of interaction with the folded domain or more specifically loop 74 – 83, which is more dynamic in both our study as well as in ref 31. This is puzzling data that needs more investigation. We are planning some mutation studies involving this region. When we get more labeled protein, we also want to lower the pH for one sample and see if our data becomes more like ref 31. Hopefully we can also get an answer when we will do actual dynamic measurements on the nsp1 protein. Having real data is of course always more interesting than predicted/simulated. As you mention, it would be exciting if this is a switch between different functions, but we feel it is too early to speculate. Our virology partners are planning their experiments at the moment and hopefully we will have some data in this autumn.

---

## [Editor Report · Decision Letter 1]

27 Oct 2021

1H, 13C and 15N resonance assignment of the SARS-CoV-2 full-length nsp1 protein and its mutants reveals its unique secondary structure features in solution.

PONE-D-21-14234R1

Dear Dr. Agback,

We’re pleased to inform you that your manuscript has been judged scientifically suitable for publication and will be formally accepted for publication once it meets all outstanding technical requirements.

Kind regards,

Oscar Millet

Academic Editor

PLOS ONE
---

## [Editor Report · Acceptance letter]

19 Nov 2021

PONE-D-21-14234R1 

^1^H, ^13^C and ^15^N resonance assignment of the SARS-CoV-2 full-length nsp1 protein and its mutants reveals its unique secondary structure features in solution. 

Dear Dr. Agback:

I'm pleased to inform you that your manuscript has been deemed suitable for publication in PLOS ONE. Congratulations! Your manuscript is now with our production department. 

Kind regards, 

on behalf of

Dr. Oscar Millet 

Academic Editor

PLOS ONE